# Deep Reinforcement Learning for Intelligent Dual-UAV Reconnaissance Mission Planning

**Xiaoru Zhao** [ORCID]**, Rennong Yang, Ying Zhang \*, Mengda Yan** [ORCID] **and Longfei Yue**

Air Traffic Control and Navigation College, Air Force Engineering University, Xi'an 710051, China; zxr_paper@163.com (X.Z.); yangrn6907@163.com (R.Y.); yanmd1@163.com (M.Y.); corresylf@163.com (L.Y.)
\* Correspondence: zzhangying1988@126.com

**Abstract:** The reconnaissance of high-value targets is prerequisite for effective operations. The recent appreciation of deep reinforcement learning (DRL) arises from its success in navigation problems, but due to the competitiveness and complexity of the military field, the applications of DRL in the military field are still unsatisfactory. In this paper, an end-to-end DRL-based intelligent reconnaissance mission planning is proposed for dual unmanned aerial vehicle (dual UAV) cooperative reconnaissance missions under high-threat and dense situations. Comprehensive consideration is given to specific mission properties and parameter requirements through the whole modelling. Firstly, the reconnaissance mission is described as a Markov decision process (MDP), and the mission planning model based on DRL is established. Secondly, the environment and UAV motion parameters are standardized to input the neural network, aiming to deduce the difficulty of algorithm convergence. According to the concrete requirements of non-reconnaissance by radars, dual-UAV cooperation and wandering reconnaissance in the mission, four reward functions with weights are designed to enhance agent understanding to the mission. To avoid sparse reward, the clip function is used to control the reward value range. Finally, considering the continuous action space of reconnaissance mission planning, the widely applicable proximal policy optimization (PPO) algorithm is used in this paper. The simulation is carried out by combining offline training and online planning. By changing the location and number of ground detection areas, from 1 to 4, the model with PPO can maintain 20% of reconnaissance proportion and a 90% mission complete rate and help the reconnaissance UAV to complete efficient path planning. It can adapt to unknown continuous high-dimensional environmental changes, is generalizable, and reflects strong intelligent planning performance.

**Keywords:** intelligent mission planning; sequential decision-making problem; reconnaissance mission; deep reinforcement learning; unmanned aerial vehicle

## 1. Introduction

The combat preparedness operational planning system, from top to bottom, is divided into strategic, theater and tactical levels, of which tactical-level combat planning is called mission planning. The mission planning is an important connection between the training of troops and actual operations [1]. A focus on air operational planning can realize the engineering of combat thoughts and processes using logical thinking and information techniques. Operational planning is generally based on the task as the center of the classification process, including target analysis, task allocation, weapon allocation, tactical maneuver planning, route planning, weapon planning and navigation planning as well as other processes. One very important category in navigation planning is reconnaissance tasks. The reconnaissance tasks require the cooperation of multiple aircraft to identify and locate high-value targets (HVT). The HVT reconnaissance mission plays an important connecting role in air intelligent mission planning, which needs to quickly reach the target airspace through the area of dense threat and wander to collect information. Therefore, combined with the satellite data, the mission can provide more accurate target information to the rear

command station so as to promote the completion of strike planning. Theoretically, the mission planning is an optimization problem with multi-constraint conditions. The solution gradually evolved from the simplex method [2], a 0–1 programming model [3] and the graph theory model to the intelligent methods. The dependence on mathematical models is gradually reduced, and the intelligent optimization algorithm is used for searching and solving by referring to biological characteristics [4–6].

In recent years, the unmanned air vehicle (UAV) is increasingly used in civilian and military operations, due to its low cost, small size, ease of operation, flexibility and other characteristics [7,8]. The mission planning of its autonomous decision-making system in complex and dynamic environments has received widespread attention from academia and industry, while the capabilities on autonomous decision-making, intelligent navigation, detection and communication are emphasized [9]. However, given the particularity of applications in military processes, UAV mission planning often requires consideration of environmental threats in order to actively generate trajectories and navigation controls [10].

Multi-UAV collaborative track planning reflects different design characteristics for specific tasks. Among them, as an important type of navigation mission, reconnaissance missions against HVTs are the premise of implementing precision strikes. The reconnaissance mission supplements the preliminary information obtained by the satellite probe and ensures the completion of the follow-up strike mission. In this paper [11], based on distributed predictive control architecture and considering the collision avoidance constraints, the local trajectory in the limited time domain is obtained according to the task requirements for the multi-UAV collaborative tracking of ground moving targets. In view of coordinated strikes, [12] emphasizes the time and space constraints of simultaneous arrival and proposes a track planning method based on multi-objective optimization. Considering the communication conditions of unmanned aerial vehicles, in order to meet the full coverage of the multi-aircraft observation area, [13] establishes multi-aircraft track planning in the obstacle avoidance environment based on the Pythagorean hodographs curve. Depending on the different mission requirements, [14] uses the synthetic structure to improve the discrete particle swarm algorithm to achieve collaborative search route planning for UAV swarms.

In a real-time and dynamic environment, path planning based on optimization algorithms consumes lots of computing time to estimate environment variables. Then, real-time requirements are difficult to achieve. The search-based intelligent optimization algorithm solves the global optimal or suboptimal solution of the complex objective function through iterative optimization. However, its essence is still random search, and each solution can only be re-searched in a static known environment (objective function and constraints) that is quite complex in time and space, and it is difficult to generalize to a dynamically changing unknown environment. Therefore, its application has certain limitations when dealing with battlefield threats and rapid planning.

With the emergence of deep reinforcement learning (DRL) technology, learning-based approaches have received high attention and widespread application. The neural network learns or fits the relationship between input and output, so that the UAV can overcome the constraints of traditional planning or control methods without considering nonlinear dynamic models. Furthermore, under non-deterministic polynomial hard (NP-hard) problems, results with minimal fitting errors or optimal decision results can be obtained with very low complexity. Additionally, after storing the mapping relationship in the form of parameters, the efficient reconnaissance task can be completed in changing environments.

Due to the improvement of computation capability, the emergence of big data technology and the development of artificial intelligence algorithms, learning-based methods have developed rapidly. Among them, deep learning achieves high-dimensional mapping, reinforcement learning achieves sequential decision-making and DRL has made remarkable achievements in the application of multi-robot control [15,16], autonomous driving [17], game playing [18] and navigation support [19]. How DRL solves the problem of navigation under different planning task styles has received widespread attention. The

authors in [20] proposed a UAV situation assessment model with real-time target and UAV location information based on deep Q-network (DQN). When considering discrete actions, Ref. [21] presented a DRL framework to accomplish some partially observable detect tasks based on deep deterministic policy gradient (DDPG), and Ref. [22] proposed the layered recurrent Q-network (layered-RQN) algorithm; this algorithm decomposes the obstacle avoidance navigation problem of the UAV, and the authors used the distributed DRL framework to gradually learn to solve the problem in highly dynamic airspace. When considering the suppression of enemy air defense mission planning, Ref. [23] established a general intelligent planning architecture based on the proximal policy optimization (PPO) algorithm, but it was not considered a dynamic situation; the experiments only involved direct strikes without situation changes during flight. Unfortunately, the UAV mission planning designed in these studies has weakened the obstacles and purpose requirements in the navigation mission planning, so there is no guarantee of the generalizability of the DRL agent or the robustness of navigation control, especially for target reconnaissance and jamming.

The above research papers include a large amount of optimized-based and learning-based methods of UAV navigation and provide a solid research foundation for UAV target reconnaissance and jamming mission planning. Taking the HVT reconnaissance task as an example, this paper proposes an efficient reconnaissance intelligent task planning method based on deep reinforcement learning in end-to-end mode. In reconnaissance mission planning, in order to effectively drive the operation, the electronic suppression jammers and the stealth reconnaissance UAV are dispatched. The two aircraft reach the target airspace cooperatively. The target information is obtained by reconnaissance. The battlefield threat situation is updated to guide the follow-up strike force and improve the operation loop. The contribution of this study includes three points as follows:

1.   First, the task planning problem of dual-UAV coordinated reconnaissance is described, and the action space and environment space are analyzed to meet MDP.
2.   Second, the principle of the DRL algorithm is introduced. According to the requirements of UAV reconnaissance for HVT missions, the clip reward function is set to reduce the influence of sparse reward, improving the algorithm convergence.
3.   Finally, based on the PPO algorithm, a dual-UAV collaborative reconnaissance with multi-radar detection threaten environment mission platform is established. The experiments evaluate the reconnaissance capability and decision-making essence and analyze the superiority and potential value of this method.

## 2. Problem Formulation

When suppressing the enemy's air defense mission during high-threat electronic warfare scenarios, the coordination of dual-UAV reconnaissance and jamming is essential, in order to improve the intelligence accuracy and promote the subsequent missions. During reconnaissance missions, the reconnaissance UAV and the jammer need to cooperate and cross the enemy radar detection area protected by HVTs, safely and rapidly. Considering the dependence of the DRL algorithm on mode, we set up a simulation environment for cooperative ground HVT detection. As shown in Figure 1, unmanned aircraft need to quickly approach the target while avoiding the surrounding radar detection. During the maneuvering process, a mathematical description consistent with MDP is established. The description can be effectively solved by DRL methods.

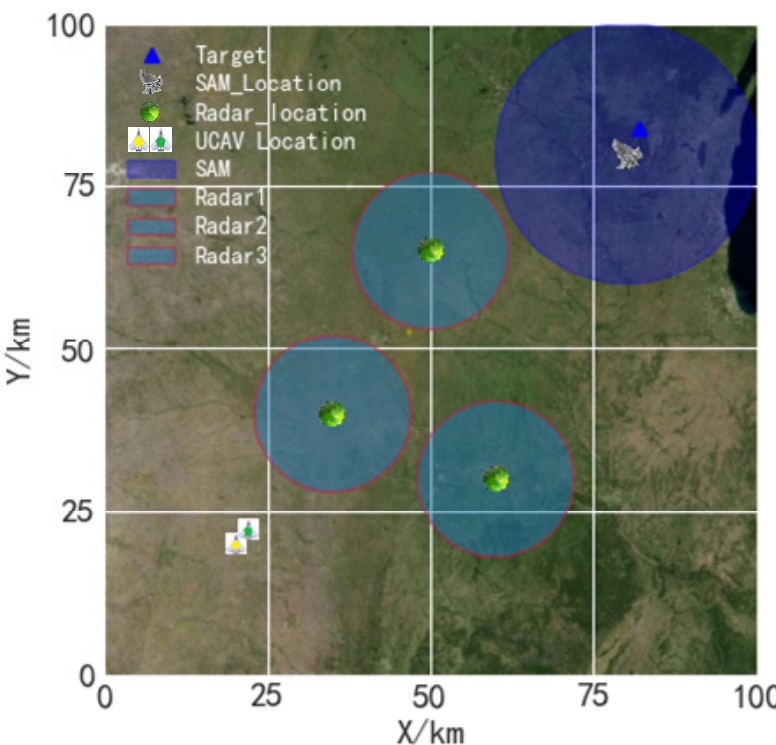

**Figure 1.** Schematic of a dual-UAV cooperative reconnaissance mission.

*2.1. Map Description*

Comparing the single maneuver distance of the aerial vehicle with the departure height of aircraft, the far-field condition is matched. Thus, the simulation scene is appropriately simplified to focus on the important reconnaissance mission target planning. For the purpose of brevity and without loss of generality, as suggested in paper [22], we assume that the UAV flies at a fixed altitude with the help of autopilots, where flight height H is a positive constant.

As shown in Figure 2, when formulating real-time tracks, owing to the constraints of aircraft mobility and dual-aircraft cooperation, the nearest distance to the radar detection range $d_{\min}$, the distance to the target $d_{target}$ and the relative distance $d_{re}$ between two UAVs need to be dynamically considered.

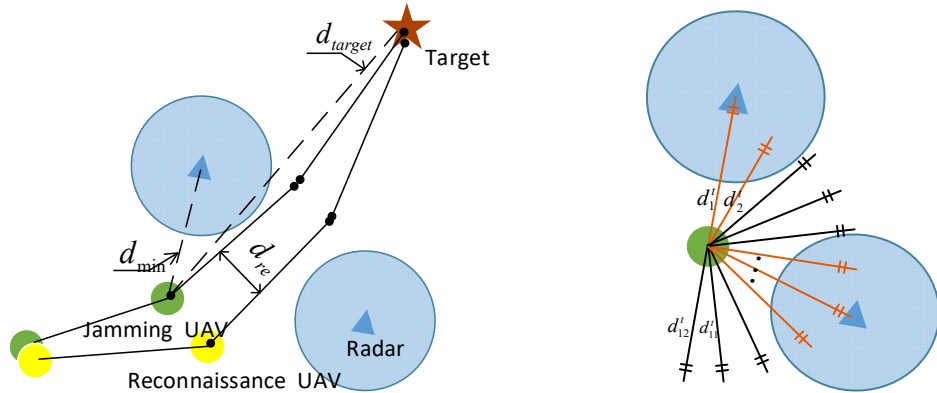

**Figure 2.** Distance diagram in the environment.

Although many inertial navigations equipped with auxiliary autopilots can provide high-precision dynamic information [24], ultrasonic rangefinders cannot meet the require-

ments of detection distance in the combat environment. Therefore, phased-array radar is selected as the distance sensor.

The UAV distance observation matrix is consisted of observation vectors in different directions provided by auxiliary sensors. As shown in Figure 3, $t$ is represented by a distance of 12 different directions, $D_t = [d_1^t, d_2^t, \ldots, d_{12}^t]$. The minimum value is obtained by comparison. If the minimum value is less than the radar detection range, it indicates that the UAV was found at this time. On the contrary, the minimum distance becomes larger if the UAV is not found.

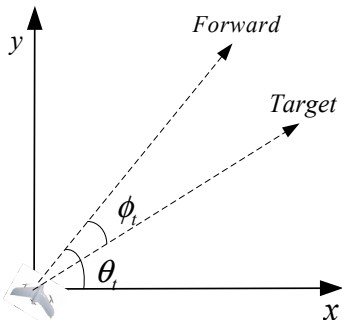

**Figure 3.** Diagram of the yaw angle and relative angle.

According to references [21,22], the earth fixed coordinate can be used to describe the relative and absolute positions of UAVs, which are expressed as the local coordinate system shown in Figure 3. With UAVs flying at a fixed altitude, the absolute position can be described by yaw angle $\theta_t$, and the relative position can be described by angle between UAV and target $\phi_t$. Above all, the range observation vector, distance between two aircraft, yaw angle and relative angle are integrated to represent the environment space in the mission planning process, $S_t = [D_t, \theta_t, \phi_t, d_r] = [d_1^t, d_2^t, \ldots, d_{12}^t, \theta_t, \phi_t, d_r]$, where $-\pi/4 \leq \theta_t, \phi_t \leq \pi/4$.

### 2.2. Kinematics of UAV

In the course of flight, the control of a UAV generally adopts a three degrees of freedom model. According to reference [25], there is strong coupling between UAV velocity and distance and weak coupling between acceleration and distance. In practice, the new generation of autopilots (such as Pixhawk) can autonomously fly according to the control signal, so that the UAV can realize autonomous flight control by providing position and speed. Therefore, for the sake of simplifying the mission planning research, we ignore the control inner loop and mainly focus on the distance and velocity information of UAVs. The kinematics equations are expressed as follows:

$$\begin{cases} x_r^{t+1} = x_r^t + v_r \cos(\Delta\theta_r^{t+1}), \\ y_r^{t+1} = y_r^t + v_r \sin(\Delta\theta_r^{t+1}), \\ x_j^{t+1} = x_j^t + v_j \cos(\Delta\theta_j^{t+1}), \\ y_j^{t+1} = y_j^t + v_j \sin(\Delta\theta_j^{t+1}), \end{cases} \tag{1}$$

where $p_r^t = [x_r^t, y_r^t]$ and $p_j^t = [x_j^t, y_j^t]$ represent the positions of the reconnaissance and jammer in the coordinate system; $v_R$ and $v_J$ denote the range of velocity as $(50, 200]m/s$; yaw angle at this time is $\Delta\theta_r^{t+1}$ and $\Delta\theta_j^{t+1}$ describes the motion vector of aircraft. To avoid sudden changes in aircraft navigation direction during training, the yaw angle is regulated as $\Delta\theta = \left|\Delta\theta_r^{t+1} - \Delta\theta_r^t\right| \leq \pi/4$.

The objective function can be written as

$$
\begin{cases}
\min \; \sum\limits_{0}^{\tau} \| \mathrm{p}_r^{t+1} + \mathrm{p}_j^{t+1} - (\mathrm{p}_r^t + \mathrm{p}_j^t) \|_2, \\
s.t. \; d_{\min} \geq d_{radar}, \; d_{re} > \overline{d}, \\
\mathrm{p}_o = [x_o, y_o], \; \mathrm{p}_{Radar} = [x_{Radar}, y_{Radar}], \; \mathrm{p}_T = [x_{target}, y_{target}],
\end{cases}
\tag{2}
$$

where, $\overline{d}$ denotes the minimum distance avoiding the cooperative collision between two UAVs; $\mathrm{p}_o$, $\mathrm{p}_{Radar}$ and $\mathrm{p}_T$ represent the starting point, radar deployment location and HVT position.

In summary, this paper establishes the state space and action space for the task planning of cooperative jamming reconnaissance and confirms that it conforms to the MDP problem. The process of solving the MDP problem through DRL is described as follows. The UAV starts from a certain state. During the flight, it makes an action based on the perception of the current environment, and it obtains an immediate reward. It enters the environment at the next moment with the transition probability $p(s_{t+1}|s_t)$, and it continues to select new actions until the game round ends. Hence, in an episode, the formation of environment, action and reward depends on the transition probability to achieve the maximum cumulative reward decision-making process as the goal. The problem is of decision optimization.

## 3. Deep Reinforcement Learning in Reconnaissance Mission Planning

In this section, the establishment of reinforcement learning framework for dual-UAV cooperative reconnaissance HVT mission is introduced.

### 3.1. Basic Principle

Reinforcement learning is usually used to solve sequential decision-making problems. A Markov decision process framework is usually established to describe the problem, which is represented by tuples $(S, A, P, R, \gamma)$, where $S$ indicates a finite state, $A$ shows a finite action set, $P$ is a state transition function, $R$ represents reward return function and $\gamma$ is used to calculate the long-term cumulative reward as discount factors. Since the traditional reinforcement learning is developed in tabular reinforcement learning, the discrete action value function is described by the limited state and action. However, for the continuous high-dimensional beam problem, there will be a 'dimension disaster' problem. Therefore, the approximate value function and strategy search of function approximation theory are proposed. With the advantage of a deep network, the DRL method is obtained, and the algorithm performance is further improved [26–29]. In accordance with policy updating and learning methods, DRL algorithms are generally divided into three categories: value function based, direct policy search based and AC based.

The goal of deep reinforcement learning is to search for parameterized policy $p_\theta(s, a) = \mathrm{P}[a|s, \theta]$ according to the direction of maximizing expected reward $J(p_\theta) = E_{\tau \sim p_\theta}[R(\tau)]$, where the reward return is expressed as the expected value of discount reward and shows as $\overline{R}(\tau) = E_{\tau \sim p_\theta}[\sum_{t=0}^{\infty} \gamma^t r_t]$ in an episode $\tau = (s_0, a_0, s_1, a_1, \cdots)$, where $\theta$ denotes a policy parameter. The obtained optimal strategy can be written as:

$$
p_\theta^* = \underset{p_\theta}{\mathrm{argmax}} E_{\tau \sim p_\theta} \left( \sum_{t=0}^{\infty} \gamma^t r_t \right)
\tag{3}
$$

The actor–critic reinforcement learning algorithm combines the first two methods, and the algorithm framework is displayed in Figure 4. The strategy network is represented as actor, and the value function network is the critic. The two network updates are guided

by the value function error, so as to accelerate the learning speed. The policy is renovated through the expected return gradient, which can be expressed as follows:

$$\nabla J(\theta) = E_{\tau \sim p_\theta} \left[ \sum_{t=0}^{T} R(\tau) \nabla_\theta \log p_\theta(a|s) \right] \tag{4}$$

where $p_\theta(a|s)$ denotes the actor network and $R(\tau)$ plays the critic role. $R(\tau)$ can also be achieved by state-behavior function $Q^\theta(s_t, a_t)$, TD residual $G_t = r_t + V^\theta(s_{t+1}) - V^\theta(s_t)$, Monte Carlo method, etc.

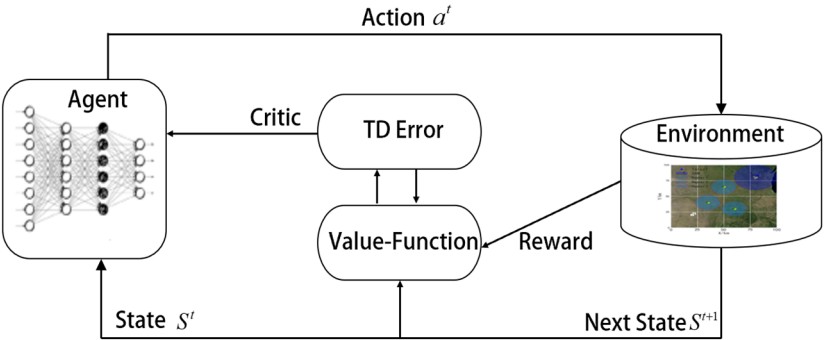

**Figure 4.** Schematic of actor–critic reinforcement learning.

### 3.2. Proximal Policy Optimization Introduction

In view of the requirement of continuous long-term flight planning for dual-aircraft cooperative reconnaissance missions, the DRL algorithm based on continuous action control is preferred. The PPO algorithm [30] is a stable, reliable and easy-to-implement actor–critic algorithm that was widely used in the training of Open AI DOTA2 agent Open AI Five [31] in 2019 and Jue Wu [32] in 2020.

The PPO algorithm approximately solves trust-region policy optimization (TRPO), inherits the stability and reliability and reduces the amount of calculation in the meantime.

The algorithm makes a first-order approximation of the objective function to ensure that the performance of the strategy increases monotonously in the optimization process and optimizes the surrogate loss function. Meanwhile, the policy is optimized in the direction of maximizing the expected return (minimizing the loss function) by limiting the distance between the new strategy and the old strategy with the clip function. In the process of the engineering implementation and debugging, the algorithm achieves a balance between sampling efficiency, algorithm performance and complexity.

With the PPO algorithm, the advantage function is used to replace the expected return in the critic part to measure the quality of the action and increase the stability of the algorithm. Equation (4) is rewritten as follows:

$$\nabla J(\theta) = E_{\tau \sim p_\theta} \left[ \sum_{t=0}^{T} A_\theta(s, a) \nabla_\theta \log p_\theta(a|s) \right] \tag{5}$$

where $A_\theta(s, a) = r(s, a) + \gamma V_\theta(s') - V_\theta(s)$ denotes the difference between the expected return from action $a$ and the previous one under environment $s$. Generalized advantage estimation [33] has been used to calculate the advantage function and keep the variance and deviation estimated through the value function.

Since PPO is an on-policy algorithm, for the purpose of improving the sample utilization rate and controlling the training direction of the new strategy, the importance sampling technology is added, and Equation (5) is changed into:

$$\nabla J(\theta) = E_{\tau \sim p_\theta} \left[ \frac{\nabla p_\theta(a_t|s_t)}{p_{\theta_k}(a_t|s_t)} A_{\theta_k}(s, a) \right] \tag{6}$$

where $p_{\theta_k}$ and $p_\theta$ represent policies that interact with the environment and are being trained, respectively. The clip function is introduced to limit the update speed of the policy while retaining the training stability. The obtained objective function is organized as:

$$J^{CLIP}(\theta) = \sum_{(s,a)} \min\left[\frac{p_\theta(a_t|s_t)}{p_{\theta_k}(a_t|s_t)} A_{\theta_k}(s,a), clip\left(\frac{p_\theta(a_t|s_t)}{p_{\theta_k}(a_t|s_t)}, 1-\varepsilon, 1+\varepsilon\right) A_{\theta_k}(s,a)\right] \quad (7)$$

where $\varepsilon$ indicates hyper parameters. The pseudo code of the algorithm is shown in Algorithm 1.

---

**Algorithm 1 Proximal Policy Optimization Algorithm (CLIP)** [30].

---

1. for $i \in \{1, 2, \ldots, N\}$ do
2.  Run policy $p_\theta$ for $T$ timesteps, collecting $\{s_t, r_t, a_t\}$
3.      Estimate advantages $A_t$
4.      Given policy parameters $\theta_{old} \leftarrow \theta$
5.      for $j \in \{1, 2, \ldots, M\}$ do
6.          Sampled from the generated trajectory
7.          Estimate policy loss function and value loss function
8.          Optimized objective function $J_{PPO}(\theta)$
9.          Update $\theta'$ based on $\nabla J_{PPO}(\theta)$
10.             $\theta_{new} \leftarrow \theta'$
11.     end for
12. end for

---

### 3.3. Environment and Reward Settings

A UAV needs to be trained to adapt to the environment and achieve the optimal control strategy for completing the mission. Environment representation and reward setting play important roles in the convergence of the PPO algorithm.

#### 3.3.1. State Representation

The completion of the jamming and reconnaissance mission in the constructed environment can effectively reduce the costs of actual testing and the incidence of accidents. However, due to the variability of the state space and the random route changes of the aircraft in the early stage of learning, the PPO algorithm needs to be preprocessed in two aspects before the state space inputs.

The first is to prevent the UAV from flying out of the boundary. During the learning process, when action $a_t$ is made, the distance to the target $d_{target} > \|p_0 - p_{target}\|_2$ indicates that it is far away from the task direction, jumping out of the loop and directly conducting the next round of learning.

Next is the standardization of the map scope, UAV position coordinates and flight parameters. Compared with the reconnaissance range of radar and target, the maneuvering distance of UAV is relatively sparse in a short period of time. Therefore, before inputting the DRL network, standardization should be carried out to accelerate the convergence. The importance of normalization is particularly essential in sparse state spaces with different unit scales. According to the definitions of the position coordinates in the previous chapter, the normalized form is expressed as follows:

$$\begin{cases} \ominus \mathrm{p}_r'^{t} = [x_r^t/L_{map}, y_r^t/L_{map}], \ominus \mathrm{p}_j'^{t} = \left[x_j^t/L_{map}, y_j^t/L_{map}\right], \\ \ominus\left(\mathrm{p}_r'^{t} - \mathrm{p}_{target}'\right) = [dx_{r,t}^t/L_{map}, dy_{r,t}^t/L_{map}], \\ \ominus\left(\mathrm{p}_j'^{t} - \mathrm{p}_{target}'\right) = \left[dx_{j,t}^t/L_{map}, dy_{j,t}^t/L_{map}\right], \\ \ominus v_R' = v_R/V_{\max}, \ominus v_J' = v_J/V_{\max}, \end{cases} \quad (8)$$

where $\ominus$ stands for the standardized processing function.

Related research shows that deep learning has batch normalization technology [34]. Nevertheless, according to the test results in [21], the method of manually ensuring the input state of DRL within the set range is more conducive to the convergence of the algorithm.

### 3.3.2. Reward Shaping

Shaping the reward function can guide the interaction between the agent and the environment, evaluate the effectiveness of the action and make the correct judgement. Hence, reasonable reward setting can effectively improve the convergence speed of the algorithm. Faced with a cooperative reconnaissance and jamming mission, the mission is divided into two stages. The first-half flight aims at avoiding radar detection and reaching the target reconnaissance airspace on the shortest path. The target of the latter flight is wandering in the target airspace based on the cooperation of the jammer and steal reconnaissance UAV.

As stated by the specific task stratification, the reward is divided into the following four parts, and to increase the convergence stability, the reward is limited to $[-1, 1]$ by CLIP function.

1.  Closing to target. The reward needs to be set to conduct the UAV to gradually approach the target in the environment exploration. Under the security distance, the closer the UAV is to target, the higher the rewards as $r_{approach}$. To speed up reaching the target, assign a penalty function $r_p$ to urge the agent to avoid meaningless wandering:

$$r_{approach} = 10 \times clip(e^{-\alpha(d_{jt}+d_{rt})}, 0, 1) \ \ r_p = -c \tag{9}$$

where $\alpha$, $c$ are constants, and $d_{jt}, d_{dt}$ can be obtained with a rangefinder device. The first-reword function is defined as $r_1^t$.

$$r_1^t = r_{approach} + r_p \tag{10}$$

2.  Avoiding radar detection. When approaching the target, UAVs need to avoid the threat of radar detection so as to plan a reasonable flight trajectory. The closer the distance to the radar, the greater the penalty:

$$r_2^t = -clip(e^{-\beta d_{\min}}, 0, 1) \tag{11}$$

Considering the detection radius of radar, $d_{\min} = \min(D_t)$ is obtained based on the distance defined in Section 2.1.

3.  Wandering reward. When the jammer approaches the target detectable airspace, the searchable radius of the ground-to-air missile force is suppressed. Simultaneously, the reconnaissance aircraft implements reconnaissance, depending on its own stealth performance, under the premise of safety, the maximum distance to the target and wandering. For the aircraft performing reconnaissance missions, the radar cross section (RCS) $\zeta$ can be reduced by relying on their own stealth coatings to reduce the probability of discovery; generally $\zeta^{0.25}$ indicates the RCS [21]:

$$r_3^t = \begin{cases} 1 - \|\mathbf{p}'^t_r - \mathbf{p}'_{target}\|_2 & if \ \rho_{\max}\zeta^{0.25} < \|\mathbf{p}'^t_r - \mathbf{p}'_{target}\|_2 < \rho_{\max}, \\ -\|\mathbf{p}'^t_r - \mathbf{p}'_{target}\|_2 & if \ \|\mathbf{p}'^t_r - \mathbf{p}'_{target}\|_2 < \rho_{\max}\zeta^{0.25}, \\ 0 & else, \end{cases} \tag{12}$$

where $\rho_{\max}\zeta^{0.25}$ represents the dynamic detection distance of the UAV by radar and $\rho_{\max}$ denotes the maximum radar range. Considering the non-idealization of the actual stealth efficiency, $\zeta^{0.25}$ is set as an attenuation value to ensure the safety of the detection unmanned air vehicle.

4. Collision avoidance. During the cooperative of two UAVs, a safe distance is set to establish avoidance penalty function:

$$r_4^t = -clip(\frac{d_s - \Delta d}{d_s}, 0, 1) \tag{13}$$

where $d_s$, $\Delta d$ indicate the safe and the relative distance between dual UAVs.

The above rewards are integrated into Equation (14),

$$R = \omega_1 r_1^t + \omega_2 r_2^t + \omega_3 r_3^t + \omega_4 r_4^t \tag{14}$$

where $\omega_1, \omega_2, \omega_3, \omega_4$ represent the weight value of the above four rewards. The flight characteristics of the agent can be changed by adjusting the weight.

Dual UAVs are to reconnaissance what agents are to the DRL algorithm. Based on the PPO algorithm, the autonomous reconnaissance mission planning framework is shown as Figure 5. The framework implements an agent's end-to-end decision-making, which consists of offline training and online planning. The training part is divided into a pair of actor networks and critic networks, where agents and the environment conduct interactive training. After the reward converges, policy is stored in the multi-path storage. The planning part integrates policy and the initial situation to make mission planning.

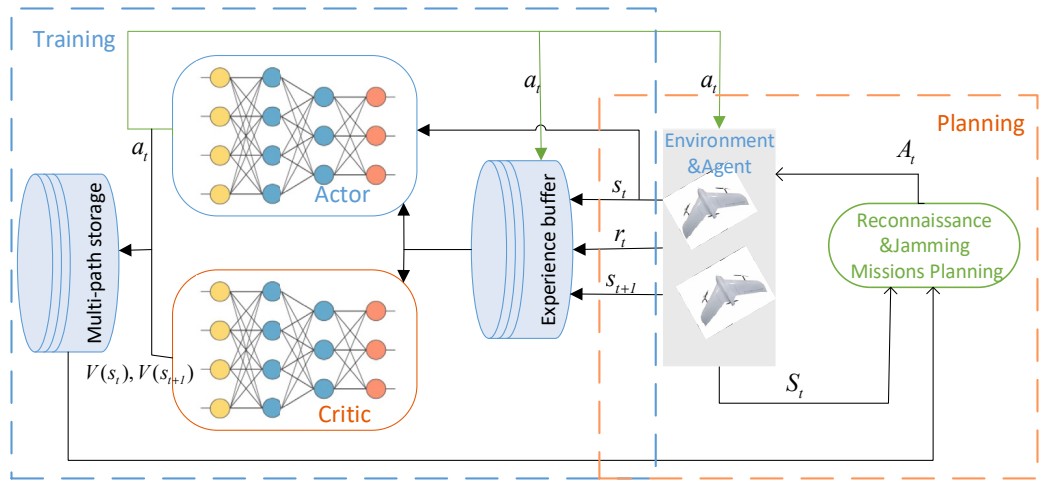

**Figure 5.** The target reconnaissance mission planning framework.

## 4. Experiment

This section describes the simulation environment and the initial hardware and software configuration. Secondly, the primary parameters of the PPO algorithm are configured. Finally, according to the evaluation indexes, we use the algorithm to test the reconnaissance mission planning at different difficulty levels and with different DRL algorithms. The corresponding results and analysis are presented in the end.

### 4.1. Experiment Establishment

The simulation sets the mission planning scenario in a square area with size 100 km × 100 km. The jammer and reconnaissance UAVs took off from coordinates (20,20) and (23,23), respectively. The two UAVs passed through three radar detection areas to reach the target protected by ground-to-missile, which is deployed in (80,80) and performed the reconnaissance task when the relative distance to the target is less than 20 km. The radar deployment coordinates are set to (35,40), (60,30) and (50,65), and the detection distance is 15 km. The essential detectable distance of the ground-to-air missile force is 20 km, which is reduced to 10 km when interference is suppressed. According to the introduction in Section 3.3.1,

the distance and coordinate in the environment are standardized in advance during the test to improve the training effect of the neural network and stabilize the convergence of the algorithm.

Since the UAV equipped with radar completes the reconnaissance task, the scanning area per unit time of the radar is related to its point scanning period. The time interval of the two UAV action plans is defined as the action response period $\tau$. According to the summary in [21], the period needs to meet the condition $\tau > \sqrt[3]{4(\zeta^{0.25}\rho_{\max})^2 \tau_r / V_{\max}^2} = 0.0737s$ where $V_{\max}$ is the speed of the aircraft and $\tau_r$ is the electrical scanning period of the radar.

The task planning for UAV cooperative reconnaissance jamming is the result of the training of the agent relying on probability reasoning and reward guidance, and the allowable execution time of each round is set to 500 s. During the planning process, when the agent is close to and maintains a period of wandering posture, or is too close to the radar detection range in the flight process, the task ends in advance. Otherwise, the plan ends after the maximum steps are completed.

In order to investigate the ability of the agent to perform mission planning in the built simulation environment, the test is carried out by changing the locations and numbers of radars in the state. Four task scenarios are set in Section 4.5, and the generalization of the algorithm is proved by comparing the evaluation indexes.

At the beginning of each episode, the UAVs and radar state information are reinitialized to ensure the generalization of the algorithm. During training, (1) the loss of advantage function and value function are standardized to increase the stability of the strategy training; (2) by adopting an adaptive learning rate and adaptive clip value, the learning rate can be maintained in the early period of training to accelerate the convergence speed and strategy update and gradually reduced at the later stage of training to maintain a stable modernization.

The simulation environment is Python 3.6 and PyCharm, and the deep learning library is implemented by Pytorch. The intelligent planning experiments of four different scenarios are completed and compared with other classical deep reinforcement learning AC algorithms. Other hyperparameters that execute simulation are set in Table 1.

**Table 1.** Hyperparameter settings.

| Parameter | Value |
| --- | --- |
| Neural Network | torch.nn.init.orthogonal_() |
| Optimizer | Adam |
| Num_episode | 600 |
| Learning Rate | 0.0003 |
| Clip | 0.2 |
| Minibatch Size | 64 |
| Num_seed | 3 |
| Step Per Round | 500 |
| Bunch Size | 2048 |
| $\alpha$ | 4 |
| $\beta$ | 2 |
| $c$ | 0.01 |
| $\zeta^{0.25}$ | 0.1 |

### 4.2. Evaluation Indicators

There are four indexes established to evaluate the simulation results:

- Mission Complete Rate (MCR): Refers to the percentage of episodes in which the reconnaissance UAV finishes by successfully evading the search and completing a reconnaissance mission. The value can be reported after offline training episodes have been run. This indicator can evaluate the learning efficiency of the evaluation environment and reward settings in the algorithm;
- Risk Times (RT): RT represents the total number of UAVs under radar detection. The indicator value is calculated by statistically online-test planning the UAVs trajectory;

- Proportion of Reconnaissance Completed (PRC): PRC is the maneuver response period of the aircraft successfully approaching the target, revealing the agent's understanding of task completion;
- Flight Length (FL): The whole flight path length from the starting to the target airspace.

### 4.3. Experiment I: Convergence

For sake of the effectiveness of the training model for cooperative reconnaissance mission planning, the average reward learning curves of different DRL algorithms are calculated. The DRL algorithm advantage actor–critic (A2C) is based on AC type, and TRPO [35] and TD3 [36] are compared. We set three different random seeds to train the strategy network and value function network. As part of recording the time step and cumulative reward of each episode, the average reward learning curve is calculated and compared, as illustrated in Figure 6. The higher final average rewards with less training duration are, the better the convergence of the method.

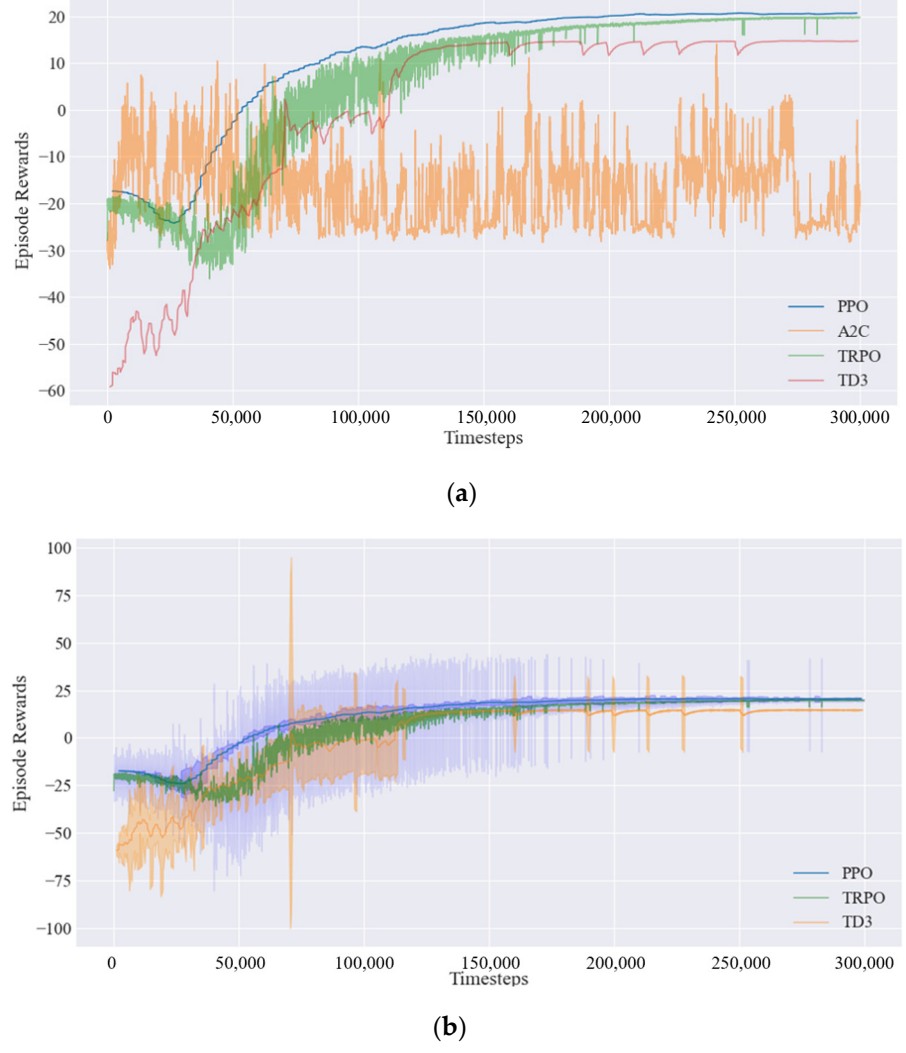

**Figure 6.** Convergence curves for the reconnaissance HVT scenario. (**a**) The convergence curves of three algorithms. (**b**) The convergence curves of two algorithms. The shaded region denotes the standard deviation of average evaluation over three trials.

As it can be observed from the above Figure 6a, PPO, TRPO and TD3 can reach their convergence point after around 16,000 and 25,000 timesteps, while A2C cannot come to a converge point. Further observation in Figure 6b explains that the rewards curve for

TRPO and TD3 still fluctuates temporarily until the later stage of training, so the stability of convergence performance is weaker than that of PPO. Compared with PPO, TRPO and TD3 achieve lower average reward value. Therefore, Figure 6 indicates that the PPO model used in this work has higher episode rewards, more stable training, smaller episode reward variance and good robustness.

### 4.4. Experiment II: Effectiveness

In accordance with our deployment of a reconnaissance aircraft and a jammer, UAVs pass through the enemy radar detection region and are close to the target airspace deployed by the ground-to-air missile. After entering the target protected aera, the two UAVs wander close to target for a period of time to complete the reconnaissance task while cooperating to ensure safety. The established intelligent planning model is used to solve the problem, and 600 rounds of offline training are carried out to obtain the policy network model. Then, 100 online plannings are executed to test the intelligent planning ability, and the completion of cooperative reconnaissance HVT mission is obtained, as shown in Figure 7.

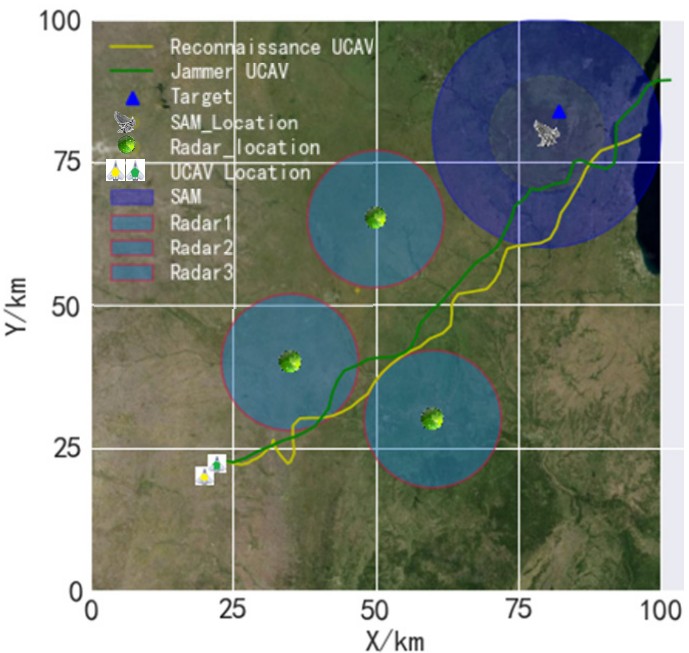

**Figure 7.** Mission planning of the dual-UAV cooperative reconnaissance high-value target.

It can be observed from Figure 7 that when the reconnaissance UAV and the jammer take off in turn and they pass through the radar detection area, the jammer can sacrifice part of the distance constraint so as to quickly cross due to its suppressing interference or enter airspace in time to cover the reconnaissance one out of dangerous space, when it enters the radar detection airspace.

When reaching the target, the jammer implements suppression interference, providing the time for the reconnaissance aircraft to complete the task. Meanwhile, the reconnaissance fighter approaches the target under the premise of ensuring security. After the task is completed, the two aircraft cooperatively fly away. In summary, the trained model has achieved strong intelligent collaborative planning ability.

### 4.5. Experiment III: Generalization

In the reconnaissance mission, increasing radar detection threat area not only increases the difficulty of UAV navigation decision-making but also puts forward a more severe test of the algorithm generalization because of the synchronous amplification of the learning sample environment.

To demonstrate the ability of efficient reconnaissance in the simulation training environment built by the PPO algorithm, we establish a comparative experiment. The location and number of warning detection radar are changed from 1 to 4, and the offline planning is accomplished for 100 episodes after the online training. The mission plannings under different situations are shown in Figure 8.

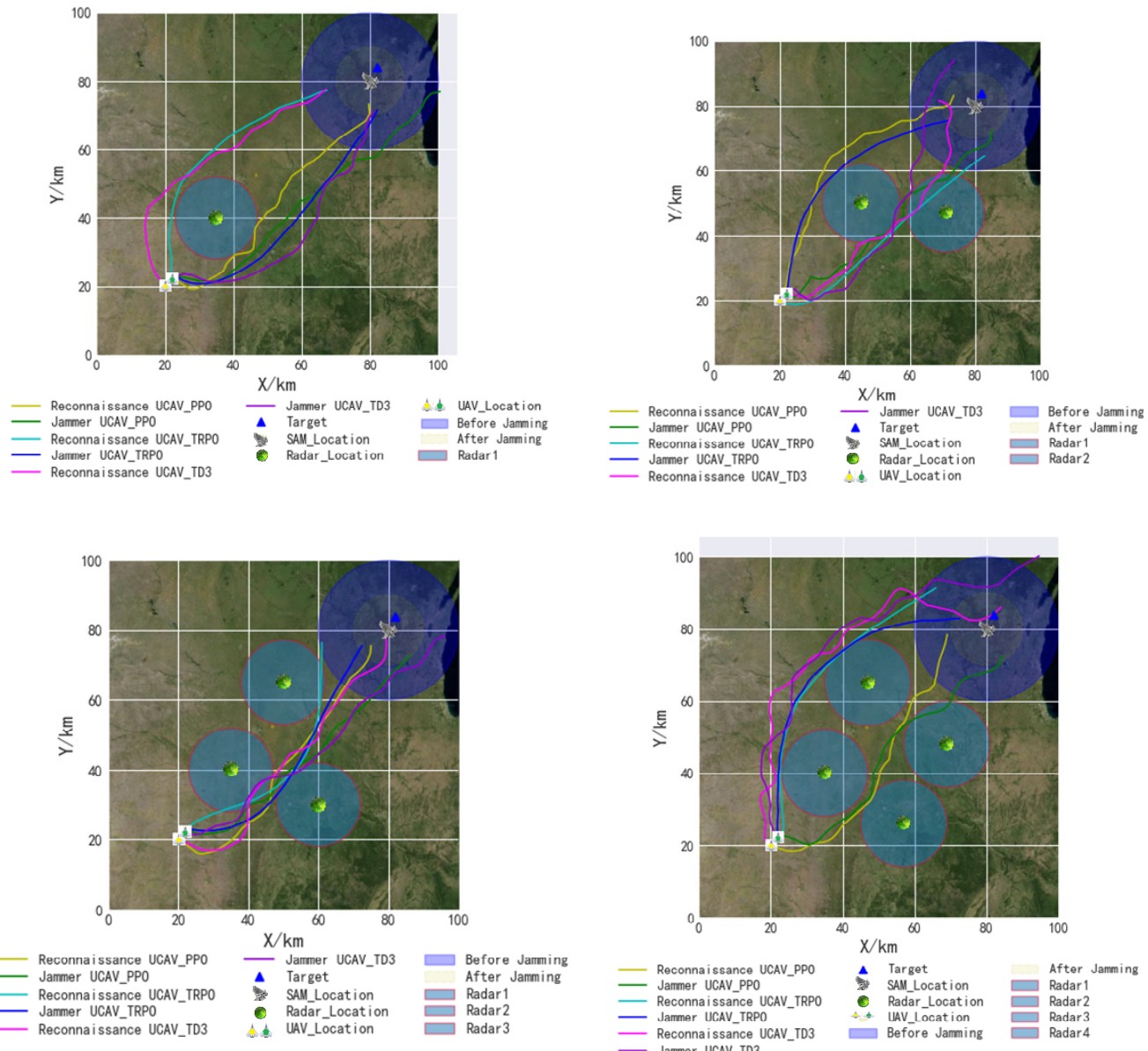

**Figure 8.** Intelligent mission planning under different situations.

As shown in Figure 8, the converged DRL methods are deployed with the increasing number of radars. It can be observed from Figure 8 that the UAVs trained by the TD3 algorithm cannot simultaneously complete the multiple targets of avoiding radar threats and the rapidly approaching target. The UAVs trained by TRPO have difficulty meeting the requirement of the fast-approaching target. Compared with TRPO and TD3, the agent based on PPO algorithm can better understand the mission setting. UAVs can quickly reach the target, avoid radar detection and wander over the target for reconnaissance. The average values of the evaluation index introduced in Section 4.2 are calculated and depicted in Figure 9.

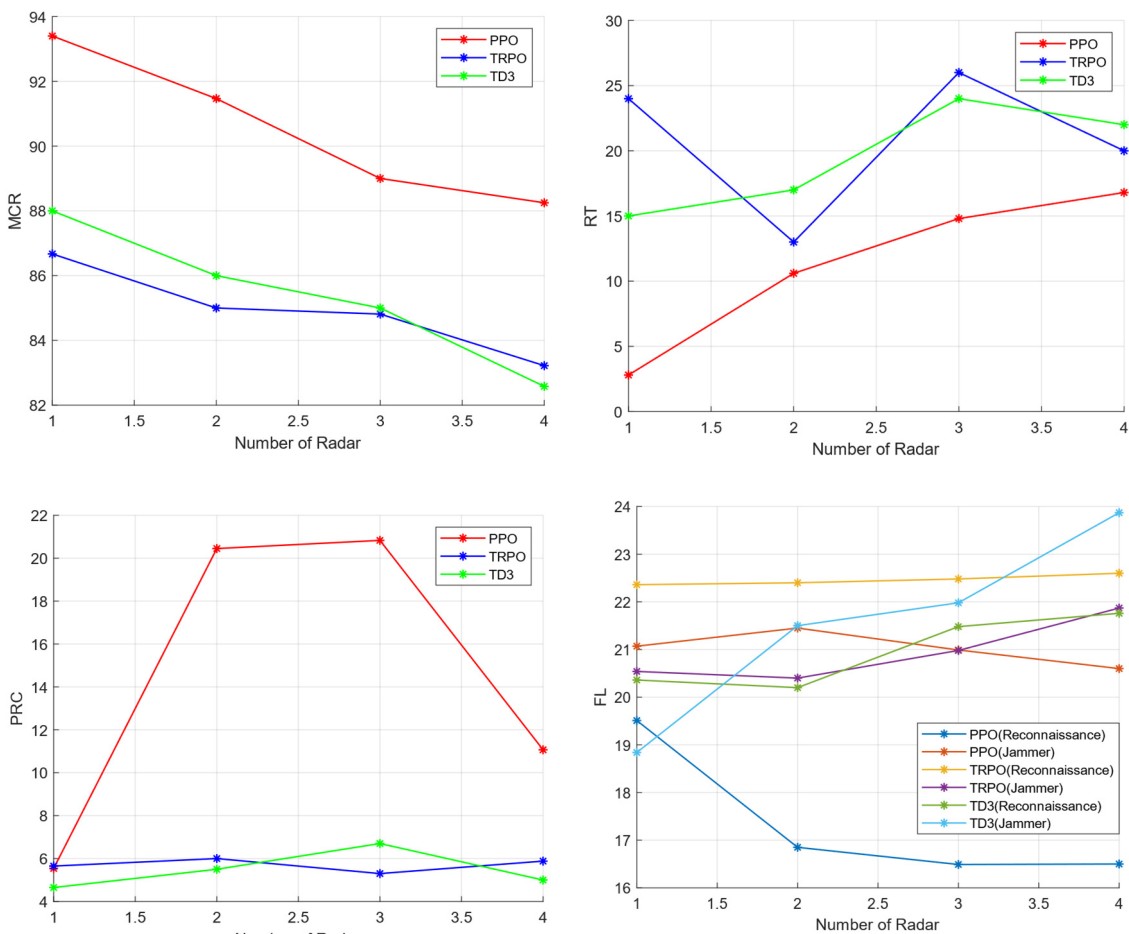

**Figure 9.** Variation curves of diverse evaluation indicators under different numbers of radars.

As illustrated in Figure 8, with the increasing number of radars, compared with the TRPO and TD3 algorithms, the transformation curves of the indexes obtained by PPO algorithm are relatively stable and ideal. In Figure 9, MCR and PRC curve changes were observed. With the increase in radar detection points, the ratio of successful learning from training convergence began to decrease, but the rate for PPO is higher than the other algorithms. Under the deployment of 2–3 radars, the offline planning performance based on PPO is more prominent, which can maintain the target airspace hovering under the condition of 20% of the total maneuver times. Under TRPO and TD3, only about 5% can be guaranteed. In summary, the two PPO evaluation indexes are always higher than TRPO and TD3.

In the process of flight, when the reconnaissance aircraft enters the radar detection range without the cover of the jammer, it is identified as being in a dangerous state. The number of maneuvering response cycles entering the threatening range is recorded, and the percentage of the total number of maneuvers is calculated, namely, the RT index. Compared with the TRPO and TD3 algorithms, when there are 3 radars, TD3 can maintain a short flight path and complete the task at low risk. However, when the number of radars increases to 4, the four indicators in Figure 9 decline significantly, and the UAV cannot complete the mission. With the increase in radar detection points, the RT value of the PPO algorithm always shows an upward trend, but the values with TRPO and TD3 fluctuate greatly. Combined with the analysis in Figure 6b, it can be seen that in the process of TRPO and TD3 offline training, even in the late learning stage, there will still be a mutation in the reward value that cannot stabilize the convergence point.

Observing the change in the FL curve, the two UAVs trained on the PPO algorithm can stabilize their flight distance with increasing radar numbers and reach the target

airspace quickly. However, the training results for the TRPO algorithm show that with the increasing complexity of the environment, the understanding of the task decreases and the flight trajectory increases significantly.

Lastly, the simulation solution proved that the intelligent planning model established based on PPO algorithm in this paper can better understand the task and has good convergence, effectiveness and generalization.

## 5. Conclusions

This paper presents an end-to-end aerial reconnaissance mission intelligent planning method. Firstly, the mission of dual-UAV cooperative HVT reconnaissance is selected as the research object, and the mission planning is described as a sequential decision-making problem. Then, the principle of deep reinforcement learning and the PPO algorithm are introduced. The state space and action space of aircraft are designed, and the intelligent planning model of jamming reconnaissance cooperative combat based on PPO algorithm is established. In order to avoid the convergence difficulty caused by sparse reward, the reward function and environment class development are designed based on clip function. In the simulation part, the offline training and online planning mode are adopted to complete the collaborative planning of the reconnaissance aircraft and the jammer. The convergence of the algorithm is proved by the reward curve. The effectiveness is shown in that the dual UAV can avoid radar search while quickly approaching the target with 3 radars deployed. There are four indexes established to verify the generalization of the intelligent planning model proposed in this paper. With the change in radar number, the proposed model can maintain 90% of MCR and always ensure the shortest FL of the reconnaissance UAV. The offline planning performance based on the PPO method can maintain the target airspace hovering under the condition of 20% of the total maneuver times with 2–3 radars deployed. It is concluded that the intelligent planning model proposed in this paper has certain scalability and advancement.

In the future, we will focus on splitting tasks through phase targets and further think about how multiple UAVs can accurately achieve specific tasks in the environment through hierarchical division and multi-agent collaboration.

**Author Contributions:** Study design X.Z., R.Y., Y.Z., M.Y. and L.Y.; conduct of the study X.Z., R.Y. and Y.Z.; data collection X.Z., M.Y. and L.Y.; methodology X.Z., M.Y. and L.Y.; software X.Z. and L.Y.; formal analysis X.Z. and M.Y.; writing—original draft preparation, X.Z. and M.Y.; writing—review and editing, X.Z., R.Y., Y.Z., M.Y. and L.Y. All authors have read and agreed to the published version of the manuscript.

**Funding:** This research received no external funding.

**Data Availability Statement:** The data presented in this study are available on request from the corresponding author. The data are not publicly available due to privacy.

**Conflicts of Interest:** The funders had no role in the design of the study; in the collection, analyses or interpretation of data; in the writing of the manuscript or in the decision to publish the results.

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
