# Peer review of "Deep Reinforcement Learning for Intelligent Dual-UAV Reconnaissance Mission Planning"

_electronics, doi:10.3390/electronics11132031_

Round 1
Reviewer 1 Report
This paper presents an end-to-end deep reinforcement learning-based intelligent reconnaissance method for intelligent reconnaissance mission planning for dual unmanned aerial vehicles (dual-UAV) cooperative reconnaissance missions under high-threat and dense situations. In general, the paper is interesting. It contains a good literature review to justify the contributions. The method and the results seem to be correct.
Some comments to improve the paper are:
*Include the paper organization at the end of the introduction.
*Ensure all acronyms and variables are defined in first use.
* Be consistent with the citation style; check, for instance, line 245.
*Line 423, include the figure number.
*Include the references of the algorithms used in the results section when they are not proposed in this paper.
*Include a recent method from the literature for comparing your algorithms.
*The language needs to be improved. Some sentences are quite long, the punctuation marks need to be checked. Sometimes there is an abuse of some words such as "which," and others that are repetitive along the text.
Reviewer 2 Report
Dear authors,
Thank you for your work! In my opinion the work is well organized with clear explanation of the problem, methodologies used and obtained results. In my opinion what is your work missing is an indepth discussion of your work and with small comparison of similar works, as your conclusion is very vague. Moreover, i think that you should highlight what can be further improved in your work.
In addition, few minor aspects in my opinion can be improved:
l.74 pythagorean probably?
l.102-104 those are new abbrevations, i believe they should be clarified the first time mentioned
Figure 1 - please annotate in your legend all map elements (also for the following similar figures)
Through out your manuscript your are using in-line "figure" mentioning in different forms. Please make it consistent with the template through your manuscript.
l.245 please fix the formatting of the in-line citations
l.276 remove &
l.313 - reward
l.395 - offline
Regards.
Reviewer 3 Report
It is recommended to justify the statements of abstract and conclusions with quantitative aspects mentioned in the main text of the manuscript.
Example:
"According to the concrete requirements of non-reconnaissance, dual-UAV cooperation and wandering reconnaissance in the mission, four reward functions are designed and adjusted by changing weight (how are the weights changed mathematically ...) to enhance agent understanding to the mission. To avoid sparse reward, the clip function is used to control the reward value range. ..."
By changing the location and number of ground detection areas, the experiment results show that the model is effective (quantitative aspect of effectiveness ...?) and stable (mathematical indicator of stability of the model ...?) , and can respond to the changes in the scene.
The similar changes need to be addressed in the conclusions.
